# Newborn Screening for Congenital Hypothyroidism in Japan

**DOI:** 10.3390/ijns7030034

**Published:** 2021-06-28

**Authors:** Kanshi Minamitani

**Affiliations:** Department of Pediatrics, Teikyo University Chiba Medical Center, Chiba 299-0111, Japan; kminami@med.teikyo-u.ac.jp; Tel.: +81-436-62-1211

**Keywords:** newborn screening, lowering of thyroid stimulating hormone screening cutoffs, thyroid dysgenesis, thyroid dyshormonogenesis, transient congenital hypothyroidism, permanent congenital hypothyroidism, delayed rise in TSH, low birth weight

## Abstract

Congenital hypothyroidism (CH) is the most common preventable cause of intellectual impairment or failure to thrive by early identification and treatment. In Japan, newborn screening programs for CH were introduced in 1979, and the clinical guidelines for newborn screening of CH were developed in 1998, revised in 2014, and are currently undergoing further revision. Newborn screening strategies are designed to detect the elevated levels of thyroid stimulating hormone (TSH) in most areas of Japan, although TSH and free thyroxine (FT4) are often measured simultaneously in some areas. Since 1987, in order not to observe the delayed rise in TSH, additional rescreening of premature neonates and low birth weight infants (<2000 g) at four weeks of life or when their body weight reaches 2500 g has been recommended, despite a normal initial newborn screening. Recently, the actual incidence of CH has doubled to approximately 1:2500 in Japan as in other countries. This increasing incidence is speculated to be mainly due to an increase in the number of mildly affected patients detected by the generalized lowering of TSH screening cutoffs and an increase in the number of preterm or low birth weight neonates at a higher risk of having CH than term infants.

## 1. Introduction

Since thyroid hormone is indispensable for myelin sheath formation during the fetal, neonatal, and infant periods, dysfunction of thyroid hormone during these periods causes irreversible intelligence impairment. Furthermore, thyroid hormone stimulates growth hormone secretion, insulin-like growth factor 1 production, and bone maturation. Therefore, insufficient thyroid hormone activity can result in failure to thrive and early osteoporosis in adulthood. Primary congenital hypothyroidism (CH) is the most common congenital endocrine disorder caused mainly by thyroid dysgenesis or thyroid dyshormonogenesis. CH can be prevented by early detection and optimal treatment, and newborn screening programs for CH have been introduced in many countries worldwide. In Japan, newborn screening programs for CH started in 1979 and have markedly improved neurologic and health outcomes [1,2,3]. The present review provides an update on newborn screening programs for CH as well as the treatment and long-term outcomes of CH in Japan.

## 2. Newborn Screening Programs for CH

Before the development of newborn screening programs for CH, primary CH was mainly diagnosed from clinical symptoms based on a 12-item checklist (persistent jaundice, constipation, umbilical hernia, poor weight gain, xerosis cutis, sluggishness, macroglossia, hoarseness, coldness of limbs, edema, dilation of posterior fontanel, and goiter) [4]. However, because these symptoms are nonspecific in the neonatal period, they were often diagnosed late or overlooked. Newborn screening for CH through an enzyme immunoassay-based thyroid stimulating hormone (TSH) measurement on a filter paper blood spot sample was introduced as a nationwide screening program in 1979, and this method was upgraded to enzyme-linked immunosorbent assay in 1987 [2,5,6]. At present, patients with CH are treated according to the guidelines of mass screening for CH by the Japanese Society for Pediatric Endocrinology and the Japanese Society for Mass Screening, which were developed in 1998 [7] and revised in 2014 [8].

An initial TSH-based screening is performed using a filter paper blood spot sample collected on days 5–7 postpartum. Neonates with a TSH level of 15–30 mIU/L in whole blood on the filter paper blood spot sample are immediately referred to a regional medical facility for closer clinical examination. Neonates with a TSH level of 10–15 mIU/L are retested for TSH using the filter paper blood spot sample. Neonates with a TSH level >10 mIU/L in the retested sample are usually subjected to close examination [8].

Currently, the age of the first visit for close examination of patients ranges from 15.8 to 18 days, with an average of 17.3 days [3].

During close examination in a medical facility, the family history of thyroid disease and mother’s history of iodine overload and medication are noted. In Japan, where iodine is abundant, dietary iodine insufficiency is rarely seen. A physical examination is performed mainly based on the abovementioned 12 items of the checklist. Serum TSH, free thyroxine (FT4), free triiodothyronine (FT3), and thyroglobulin levels are measured. The distal femoral epiphyseal ossification center (DFEC) is examined using X-ray, and the thyroid gland is identified using ultrasonography. Thyroid scintigraphy is reliable for the definitive diagnosis of thyroid dysgenesis. However, it is generally not performed in the neonatal period in Japan, probably because it is the only atomic-bombed country in the world.

Treatment is immediately initiated under following conditions: if a case has clinical symptoms, if the appearance of the DFEC is delayed, if the thyroid gland cannot be identified by ultrasonography, or if goiter is found. It is recommended to start treatment if the serum TSH level is ≥30 mIU/L or 15–30 mIU/L and the FT4 level is ≤15 pmol/L.

If no clinical symptoms are found, the serum FT4 level is within the normal range, and the TSH level is <15 mIU/L, a thyroid function test should be performed again. If the serum TSH level is >10 mIU/L at 3–4 weeks after birth, treatment initiation should be considered. It has been suggested that infants with a TSH level of ≥10 mIU/L at <6 months after birth and ≥5 mIU/L at 12 months after birth should be followed up carefully and treated.

## 3. Incidence of CH in Japan

Prior to the introduction of newborn screening for CH, the incidence of primary CH was 1:7400 [1]. However, once screening was started, the incidence increased to 1:3000 to 4000 since the 1990s and then to 1:2000 to 2500 since the 2000s [5]. The possible reasons for this increase include an increase in the number of mildly affected patients detected by the generalized lowering of TSH screening cutoffs and an increase in the number of preterm or low birth weight neonates at a higher risk of having CH than term infants, as well as epigenetic factors and changes in iodine intake and dietary habits [9].

The percentage of regions with the positive criterion of a TSH level of ≤30 mIU/L in whole blood on the filter paper blood spot sample doubled from 43.1% in 1993 to 89.4% in 2008 [10]. Therefore, the identification of an additional mild form of CH with gland-in-situ is thought to be responsible for the increase in CH incidence.

Preterm or low birth weight neonates are at a higher risk of having CH than term infants. In Japan, recent dramatic advances in neonatal care have led to an increase in the percentage of low birth weight neonates in Japan, i.e., from 5.2% in 1975 to 9.4% in 2017 [11]. However, there are no reports on the actual incidence of CH in low birth weight neonates in Japan.

## 4. Epidemiology of CH in Japan

Iodine deficiency is rare in Japan, which is originally an iodine-sufficient area. According to data from 1989, CH was caused by thyroid dysgenesis in 84% of cases (ectopic thyroid gland in 60% and athyreosis/hypoplasia of the thyroid gland in 24%) and by intrinsic defects of thyroid hormone synthesis (dyshormonogenesis) in the remaining 16% cases [12]. However, several recent studies using lower cutoff points for TSH levels have reported an increased diagnosis of cases with gland-in-situ. In a 2008 Japanese study, 54% of primary CH cases were caused by thyroid dysgenesis (ectopic thyroid gland in 37% and athyreosis/hypoplasia of the thyroid gland in 17%) and the remaining 46% of CH cases occurred due to dyshormonogenesis [13].

A comprehensive genetic analysis identifies genetic abnormalities in 20% of Japanese patients [14,15,16,17,18]. Mutations in the DUOX2 gene are particularly common, identified in approximately 20% cases of dyshormonogenesis. In contrast, in thyroid dysgenesis, genetic mutations can only be identified in 5–10% of patients.

Currently, the National Center for Child Health and Development (Tokyo, Japan) analyzes genetic mutations in CH-associated genes, including *DUOX2*, *DUOXA2*, *FOXE1*, *GLIS3*, *IGSF1*, *IYD*, *NKX2-1*, *PAX8*, *SECISBP2*, *SLC26A4*, *SLC5A5*, *TG*, *THRA*, *THRB*, *TPO*, *TRH*, *TRHR*, *TSHB*, and *TSHR*, using next-generation sequencing methods.

## 5. CH in Low Birth Weight Neonates

Premature and low birth weight neonates may present with hypothyroxinemia without an increase in the TSH level through a variety of mechanisms, including the hypothalamic–pituitary–thyroid axis immaturity, nonthyroidal illness, dopamine administration, high-dose steroid therapy, undernutrition, and exchange transfusion [19]. A delayed rise in TSH is a condition in which although the TSH level is below the cutoff point at initial screening, it increases later. A delayed rise in TSH is particularly common in low birth weight infants. A retrospective single-center matched case-control study shows that the percentage of small-for-gestational age infants was significantly higher in the delayed TSH rise group (71%) than in the comparison group (25%) [20]. In order not to overlook this pattern of delayed rise in TSH, since 1987, additional rescreening of premature neonates and low birth weight infants (<2000 g) at four weeks of life, when their body weight reaches 2500 g, or at discharge from the hospital is recommended, despite a normal initial newborn screening [21,22].

More than 50% of low birth weight infants before 30 weeks of gestational age manifest a temporary pattern of low levels of FT4 and normal or low levels of TSH termed ‘transient hypothyroxinemia of prematurity’ (THOP), due to the immaturity of the hypothalamic-pituitary-thyroid axis, iodine deficiency, the withdrawal of maternal placenta FT4 transfer, nonthyroidal illness and exposure to some medications. The more premature the infants are, the more severely the thyroxine is reduced. Many studies have shown that levothyroxine sodium (LT4) has a poor effect on severe hypothyroxinemia [23,24,25], and the administration of LT4 to premature infants in Japan has been suggested to cause late onset circulatory collapse [26,27]. Infants with THOP should not be treated with LT4.

## 6. Treatment of CH

The Japanese Guidelines classify serum FT4 level <5, 5 to <10, and 10 to <15 pmol/L as indicating most severe, severe, and moderate cases, respectively, taking into consideration the consensus guidelines of the European Society for Pediatric Endocrinology [28].

Treatment starts with the administration of 10 μg/kg/day LT4 in powder form once daily before breakfast. In most severe cases, treatment starts with a dose of 12–15 μg/kg/day LT4. Infants with subclinical CH can be treated with 3–5 μg/kg/day LT4 because they often become hyperthyroid when given 10 μg/kg/day LT4 [8].

The target for serum FT4 levels should be >50% of the reference range by age. The target for TSH level should be the reference range by age. A follow-up is required at one, two, and four weeks after the start of LT4 treatment, at one-month intervals until one year of age, and then at 3–4-month intervals until the adult stage.

## 7. Re-Evaluation

A re-evaluation or definitive diagnosis should be made for the patients with CH after the age of three years, including the differentiation of transient from persistent CH [8]. After four weeks of LT4 withdrawal, a ^123^I thyroid scintigram, ^123^I uptake rate, saliva/serum iodine ratio, perchlorate discharge test, thyroid function tests (TSH, FT4, FT3, and thyroglobulin), and thyroid ultrasonography are performed to diagnose athyreosis, hypoplasia, ectopic thyroid gland, hormone organification defect, and iodine concentration deficiency. If no abnormalities are detected upon these examinations, the patient is diagnosed with transient hypothyroidism. Infants treated with less than 1.25 μg/kg/day LT4 at three years of age are more likely to have transient CH [29,30]. In addition, infants who do not require an increase in LT4 dose after three years of age are more likely to have transient CH [31].

## 8. Psychomotor Development

Prior to newborn screening for CH, only 19.8% of infants with CH received treatment at an age of less than three months. Therefore, even after treatment, 43% of the patients showed mental retardation with IQ levels below 75, 33.3% of the patients showed IQ levels over 90, and two thirds of the patients were mentally retarded, including those on the borderline [32].

In early newborn screening, the recommended initial dose of LT4 was 5–8 μg/kg/day, and the initiation of treatment was often delayed until 4–5 weeks after birth. The patients with CH had an IQ that was lower by 6–20 points in comparison with controls, and the prognosis was particularly poor in severe children with a blood T4 level < 5 μg/dL at their initial visit. The mean IQ in the first nationwide survey in 1991 was 97.5 ± 14.8 (*n* = 81) [33] and that in the second survey in 1994 was 99.9 ± 13.7 (*n* = 151) [34].

Since the late 1990s, infants with CH have been treated with an initial dose of 10–15 μg/kg/day LT4, with treatment starting within two weeks after birth [35]. In the nationwide follow-up survey of CH children in 2003 [36], the DQ/IQ at 1–5 years of age was good, ranging from 104.1 to 107.3. Serious intellectual disability due to CH has almost been eradicated. However, children with severe hypothyroidism, such as athyreosis, during pregnancy presented with significantly lower IQ levels than those with other types of CH [37]. Furthermore, patients with severe CH also have cognitive, behavioral, and attention deficits in adolescence and adulthood [35,36].

## 9. Growth, Puberty, Body Composition, and Quality of Life

Prior to newborn screening for CH, while the frequency of children with a high degree of short stature equal to or less than −3 SD decreased from 45% to 11.8% after LT4 treatment, the frequency of children with a short stature equal to or less than −2 SD represented approximately 30% [32].

A report analyzing the height and body weight of 2341 patients with CH (1030 males and 1311 females) detected neither short stature nor obesity, but normal growth and constitution through a registration in the Medical Aid Program for Chronic Pediatric Disease of Specified Categories in 2002 [38]. A follow-up study in 2006 reported that the patients with CH had a nearly normal physique, with a height of 162.9 ± 8.4 cm and body weight of 60.8 ± 14.3 kg for male adults, height of 157.3 ± 5.2 cm and body weight of 52.4 ± 7.4 kg for female adults, and BMI of approximately 21.1 ± 3.0 for both males and females [39]. A report from Kanagawa shows no significant difference in adolescent growth patterns and adult height between patients with CH and healthy individuals, and no significant correlation between adult height and severity of hypothyroidism or the age of starting treatment was observed [40]. Some reports suggest that, even with good control, puberty tends to be earlier in girls with CH, judging from the age at menarche [41,42].

Patients in whom CH was detected shortly after the introduction of newborn screening have already finished compulsory education and reached the age for employment or marriage. The long-term quality of life (QOL) condition of these patients has been reported [39,41]. Regarding the employment status of these patients, full time employees represent 27% of patients, part-time employees represent 10%, unemployed represent 8%, married and unemployed represent 6%, students represent 43%, and others represent 6%, with no difference in employment status compared with the general population of the same generation. Patients with CH show no differences in academic backgrounds for employees and the unemployed compared with those of the general population, as 15% of patients graduated from university/college, 7% dropped out from university/college, 22% graduated from vocational school or junior college, 41% graduated from high school, 4% dropped out from high school, and 11% graduated from junior high school. In terms of marital status, 8% of the patients with CH are married. In Japan, more than 90% of all households purchase life insurance. Life insurance aims to cover the loss associated with life, accidents, and sickness and also meets various needs, such as savings and post-retirement security, but people with underlying conditions are often refused enrollment by life insurance companies. Among patients with CH, 46% have purchased life insurance and 65% of them applied for their insurance without declaring their disease [39].

## 10. Summary

Newborn screening for CH markedly improves the long-term intellectual outcome, physical growth, and QOL of patients with CH.

The incidence of CH is increasing every year. It is important to minimize the damage of hypothyroidism and further improve the outcomes by setting appropriate cutoff values, appropriate initial therapeutic doses, and appropriate treatment for mild CH and low birth weight infants.

Given that some patients with CH are anxious about explaining their disease to their spouses and the inheritance of the disease, proper counseling needs to be provided based on genetic diagnosis. In addition, it is becoming apparent that patients have various issues, including their transition from the pediatric to adult clinic, purchase of life insurance, and burden of medical expenses.

## Data Availability

No new data were created or analyzed in this study. Data sharing is not applicable to this article.

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
