# Peer review of "Newborn Screening for Congenital Hypothyroidism in Japan"

_2409-515X, 2021, doi:10.3390/ijns7030034_

Round 1

Reviewer 1 Report

line 80, should the line read "incidence increased" rather than "incidence reduced". lines 126-128 my need re-writing in clearer English. Line 205-210  citation required

Author Response

I would like to thank the reviewer for carefully reading my manuscript and providing me with useful comments and suggestions. I have modified my manuscript regarding their comments. Please note that the modified parts are highlighted using the Track Changes function in the revised version of the manuscript. The manuscript has been edited by a native speaker of English from a company providing professional English editing services.

Comment 1

line 80, should the line read "incidence increased" rather than "incidence reduced".

Response

Thank you for bringing this point to my attention. The notation has been revised as follows. However, once screening was started, the incidence increased to 1:3,000 to 4,000 since the 1990s and then to 1:2,000 to 2,500 since the 2000s.

Comment 2

lines 126-128 my need re-writing in clearer English.

Response

The reviewer has commented on the lack of clarity concerning in my description of transient hypothyroxinemia of prematurity. I realize that my original explanation was unclear and have revised it for clarity. The revised version is as follows. More than 50% of low birth weight infants before 30 weeks of gestational age manifest a temporary pattern of low levels of FT4 and normal or low levels of TSH termed ‘transient hypothyroxinemia of prematurity’ (THOP), due to the immaturity of the hypothalamic-pituitary-thyroid axis, iodine deficiency, the withdrawal of maternal placental FT4 transfer, nonthyroidal illness and exposure to some medications. The more premature the infants are, the more severely the thyroxine is reduced.

Comment 3

Line 205-210 citation required.

Response

I have provided the cited reference to support the explanation. There is as follows: Sato 2009.

Reviewer 2 Report

The reviewed manuscript presents long-term results of the NS CH program in Japan since 1979. Unlike other similar works, the work includes not only the results of NS CH itself, but also definitive diagnosis, treatment, reassessment of suspected cases, evaluation of transient hypothyroxinemia in premature children (THOP). The development of LT4 replacement therapy and the achieved results of the evaluation of psychosomatic development in adolescence and adulthood are also presented. From this point of view, the article is different from other works on NS CH, focused on the screening itself as on the long-term results of treatment. The interpretation of the increase in the incidence of CH over the years, attributed to the increase in prematurity and the decrease in the cutoff limit, is debatable. Other epigenetic factors deserve attention - diet (strumigens - vegetables of the genus Brassica,), iodine block of thyrosynthesis (Wolf-Chaikoff effect), etc. The contribution of the article is overview.     

Author Response

I would like to thank the reviewer for their helpful comments and suggestions. I have modified my manuscript regarding their comments. Please note that the modified parts are highlighted using the Track Changes function in the revised version of the manuscript.

Comments

The reviewed manuscript presents long-term results of the NS CH program in Japan since 1979. Unlike other similar works, the work includes not only the results of NS CH itself, but also definitive diagnosis, treatment, reassessment of suspected cases, evaluation of transient hypothyroxinemia in premature children (THOP). The development of LT4 replacement therapy and the achieved results of the evaluation of psychosomatic development in adolescence and adulthood are also presented. From this point of view, the article is different from other works on NS CH, focused on the screening itself as on the long-term results of treatment. The interpretation of the increase in the incidence of CH over the years, attributed to the increase in prematurity and the decrease in the cutoff limit, is debatable. Other epigenetic factors deserve attention - diet (strumigens - vegetables of the genus Brassica,), iodine block of thyrosynthesis (Wolf-Chaikoff effect), etc. The contribution of the article is overview.

Response

Thank you for your helpful and constructive comments. I agree that the reasons for the increase in CH incidence over the years remain to be resolved. I accept the reviewer's valuable comments that the reasons include the effects of epigenetic factors, iodine intake, and dietary habits. To address these concerns, I have made the following changes to the manuscript in line 84. The possible reasons for this increase include an increase in the number of mildly affected patients detected by the generalized lowering of TSH screening cutoffs and an increase in the number of preterm or low birth weight neonates at a higher risk of having CH than term infants, as well as epigenetic factors and changes in iodine intake and dietary habits.